# Polymeric Nanoparticles for Delivery of Natural Bioactive Agents: Recent Advances and Challenges

**DOI:** 10.3390/polym15051123

**Published:** 2023-02-23

**Authors:** Mohammed Elmowafy, Khaled Shalaby, Mohammed H. Elkomy, Omar Awad Alsaidan, Hesham A. M. Gomaa, Mohamed A. Abdelgawad, Ehab M. Mostafa

**Affiliations:** 1Department of Pharmaceutics, College of Pharmacy, Jouf University, Sakaka P.O. Box 2014, Saudi Arabia; 2Department of Pharmacology, College of Pharmacy, Jouf University, Sakaka P.O. Box 2014, Saudi Arabia; 3Department of Pharmaceutical Chemistry, College of Pharmacy, Jouf University, Sakaka P.O. Box 2014, Saudi Arabia; 4Department of Pharmacognosy, College of Pharmacy, Jouf University, Sakaka P.O. Box 2014, Saudi Arabia

**Keywords:** polymeric nanoparticles, natural bioactive agents, nanocarriers, drug delivery

## Abstract

In the last few decades, several natural bioactive agents have been widely utilized in the treatment and prevention of many diseases owing to their unique and versatile therapeutic effects, including antioxidant, anti-inflammatory, anticancer, and neuroprotective action. However, their poor aqueous solubility, poor bioavailability, low GIT stability, extensive metabolism as well as short duration of action are the most shortfalls hampering their biomedical/pharmaceutical applications. Different drug delivery platforms have developed in this regard, and a captivating tool of this has been the fabrication of nanocarriers. In particular, polymeric nanoparticles were reported to offer proficient delivery of various natural bioactive agents with good entrapment potential and stability, an efficiently controlled release, improved bioavailability, and fascinating therapeutic efficacy. In addition, surface decoration and polymer functionalization have opened the door to improving the characteristics of polymeric nanoparticles and alleviating the reported toxicity. Herein, a review of the state of knowledge on polymeric nanoparticles loaded with natural bioactive agents is presented. The review focuses on frequently used polymeric materials and their corresponding methods of fabrication, the needs of such systems for natural bioactive agents, polymeric nanoparticles loaded with natural bioactive agents in the literature, and the potential role of polymer functionalization, hybrid systems, and stimuli-responsive systems in overcoming most of the system drawbacks. This exploration may offer a thorough idea of viewing the polymeric nanoparticles as a potential candidate for the delivery of natural bioactive agents as well as the challenges and the combating tools used to overcome any hurdles.

## 1. Introduction

Natural products, such as plants, have been considered one of the major treatments for current diseases. Phytochemicals are substances obtained from herbal plants and are commonly used for non-nutritive benefits to the plants, such as protection from external dangers such as pollution, pressure, deficiency, UV, and microbial threats [1]. Out-of-date familiarity and updated findings verify comparable protective effects in humans. In particular, different phytochemicals have been well established for their antioxidants, anti-inflammatories (such as polyphenols, flavonoids, carotenoids, and allyl sulfide-containing compounds), hormones (such as isoflavones of soy), enzyme-regulating materials (protease inhibitors and indole-containing compounds), anti-infective agents (proanthocyanidins), and DNA replication modulators (saponins and capsaicin). Hence, attention has been paid to phytochemicals, particularly in pharmaceutical technology, to fabricate such compounds in easily administered and effective formulations.

However, phytochemicals suffer from several drawbacks that limit their applications in crude forms or at least lead to unambitious effects. These drawbacks include hydrophobicity, poor stability, reduced absorption and bioavailability, extensive metabolism, and fast excretion, leading to issued therapeutic effects, which can probably be overwhelmed by using nanotechnology [2].

With the alarming rise of illnesses, appropriate drug delivery has received much attention for treating and controlling diseases. Nanocarriers have been considered promising platforms for drug delivery that can enhance their therapeutic effect and overcome hurdles. They have offered an improved solubility and bioavailability of sparingly soluble drugs, enhanced preventive and pharmacological effects, and reduced adverse effects. They include nanospheres, nanocapsules, nanoemulsions, nanoliposomes, and nanoniosomes [3]. Polymeric nanocarriers (Figure 1) are manifested by their distinctive physicochemical structures, including solid polymeric nanoparticles, polymeric micelle, polymer conjugates, dendrimers, polymersomes, polyplexes, and lipomers [4]. In the last few decades, polymeric nanoparticles have received much attention and held much importance in the horizon of drug delivery. They were recognized as colloidal systems with particle sizes ranging from 1–1000 nm [5].

They are fabricated by natural, synthetic, or semisynthetic polymers. Polymeric nanocarriers synthesized from biodegradable and biocompatible polymers offer motivating preferences for controlled and targeted drug delivery [5,6,7]. Natural polymers are a concern due to their lower toxicity when compared to synthetic polymers [8], as natural polymers are biocompatible, biodegradable, safe, greatly stable, economical, and easily handled. In general, polymeric nanoparticles possess several merits (Figure 2), such as encapsulation of differently characterized actives, protection of the encapsulated drugs, and the possibility of targeting.

In this work, we have attempted to present a comprehensive review of utilizing polymeric nanoparticulate systems as promising carriers for natural bioactive agents with modern advances and persistent challenges for their effective delivery. So, this paper reviewed the formulation of polymeric nanoparticles loaded with natural bioactive agents used to treat and control different diseases. This review does not include the details of the fabrication techniques. However, it offers the principal approach to delivering natural bioactive agents via polymeric nanoparticles. To the best of our knowledge, this review addresses this essential topic, and therefore, it will assist the researchers in appreciating and formulating promising polymeric-based nanoparticles for loading natural bioactive agents by realizing the merits and challenges of the system, which could eventually be utilized in industrial development. Hence, the following aspects were considered: (a) an overview of various components of polymeric nanoparticles and methods for fabrication, (b) polymeric nanoparticles loaded with natural bioactive agents in the literature, and (c) the potential role of surface decoration and polymer functionalization in overcoming most of the drawbacks of these systems.

## 2. Components and Formulation Attributes

Principally, polymeric nanoparticles are simply composed of polymer(s), surface active agent(s), and aqueous phase. Polymers are the backbone of polymeric nanoparticles. Generally, they are made up of large repeating units designated as monomers. Currently, utilizing nanosystems for drug delivery has been fortified by the improvement/modification of actives’ properties, such as solubility, targetability, release pattern t_0.5_, pharmacokinetics, and pharmacodynamics. Accordingly, polymers are considered the cornerstone when formulating polymeric nanoparticles. Hence, it is critical for the researcher to be aware of the polymer properties, such as biocompatibility, biodegradability, stability, drug–polymer interaction, transition temperature, permeability, and safety, before starting formulation. In order to modulate the characteristics (such as release behavior, targetability, and biocompatibility) of the generated system, the functionalization of the polymer is the point of view. Changes in the polymer can be achieved by chemical modification, surface addition of targeting moiety, and incorporation of lipids to achieve the objectives of the researchers [9]. 

However, polymers are basically categorized into two classes according to their origin source; natural polymers and synthetic polymers. According to the process of development and components, the structural organization varies (Figure 3). In the case of nanocapsules, active agents might be encapsulated in the core or adsorbed at the polymeric surface, while in the case of nanospheres, active agents might be encapsulated in the matrix of the nanosphere or the surface may be adsorbed at the polymeric surface [10].

### 2.1. Natural Polymers

In the literature, there are many natural polymers frequently utilized for the development of polymeric nanoparticles, such as chitosan, alginate, gelatin, and albumin (Table 1).

#### 2.1.1. Chitosan

Chitosan (Figure 4) is a natural polymer obtained by the deacetylation of chitin. It is obtained from different natural sources such as marines, insects, fungi, or animals. In the last few decades, it has gained attention as a drug delivery platform due to its excellent biocompatibility and biodegradability [39]. Chitosan is not soluble in neutral aqueous solutions, but it can be inherently dissolved in diluted acidic solutions due to the protonation of free amine groups. Consequently, it is soluble in diluted acetic, formic, citric, and other acids [40]. On the other hand, amine groups also affect chitosan applications in the pharmaceutical field as they particularly influence mucoadhesion, permeation enhancement, transfection, and in situ gelation of chitosan [39,41]. In general, chitosan has different promising biomedical properties, for instance, as a penetration enhancer in both paracellular (by opening the tight junctions) and transcellular transport and as a mucoadhesive, with additional wound-healing activities. Several methods are utilized to prepare chitosan nanoparticles, such as ionic gelation, microemulsion, and emulsification solvent diffusion. The main advantage of the reported methods is that they use much smaller amounts of organic solvent. The most popular techniques are mentioned below: 

##### Ionic Gelation

Ionic gelation is a simple and widespread technique used for nanoparticle preparation. In this method, chitosan solution (polycationic) is prepared by dissolving in a diluted acidic solution, such as acetic acid, and thoroughly mixing (under mechanical stirring) with a polyanionic agent, such as tripolyphosphate, as a cross-linking agent leading to complexation between positive and negative charged ions and the formation of rounded shape particles. This technique is favored due to the lack of organic solvent and minimal opportunity to change the chemistry of the entrapped drug. However, this method is characterized by its poor stability in acidic environments and uneasiness in encapsulating high molecular weight actives [42,43].

##### Microemulsion

Herein, chitosan solution and glutaraldehyde are first mixed. Then, the mixture is added to a surfactant in an organic solvent. The resultant mixture is incessantly stirred until cross-linking is accomplished. After that, the removal of organic solvent can be carried out by evaporation. If desirable, the removal of extra surfactant can be carried out by centrifugation after adding calcium chloride to precipitate the excess surfactant. The preparation of nanoparticles by this method leads to the production of small particle sizes [44]. On the other hand, the incorporation of glutaraldehyde and organic solvent, as well as a longer period of processing, are the main drawbacks of this method [42].

##### Emulsification Solvent Diffusion

In this method, the preparation of oil in water emulsion is accomplished by mixing the drug (dissolved in organic solvent) solution stabilizer containing chitosan solution with continuous stirring. The resultant dispersion is further homogenized by high-pressure homogenization [45,46]. This method is favored for hydrophobic drugs as it results in high encapsulation efficiency. On the other hand, the incorporation of organic solvents and high shear forces are the main drawbacks of this method. 

##### Coprecipitation

In this method, acidic chitosan solution is added to a high pH 8.5–9.0 solution (NH_4_OH), leading to the coprecipitation of nanoparticles. This method results in the production of highly homogenous small-sized nanoparticles with high entrapment efficiency [47,48]. 

#### 2.1.2. Alginate

Alginate is an anionic water-soluble natural polymer. Chemically (Figure 4), it is 1–4 linked α-L-guluronic acid (G) and β-D-mannuronic acid (M). Owing to its biodegradability, biocompatibility, and mucoadhesion properties, it has gained attention as a drug delivery platform in the pharmaceutical field [49,50]. Mucoadhesive properties are principally attributed to the formation of strong hydrogen bonds with mucin glycoproteins via carboxyl hydroxyl interactions [51]. Generally, alginate nanoparticles are prepared by several techniques, such as ionic gelation, emulsion, covalent cross-linking, emulsification solvent displacement, and emulsion-solvent evaporation. The ionic gelation method is a broadly used technique to prepare alginate nanoparticles [52]. Ionic gelation of alginate was carried out due to its affinity towards multivalent cations such as Ca^2+^. This affinity leads to the formation of ionically or physically cross-linked alginate [50,53].

#### 2.1.3. Gelatin

Gelatin is a protein obtained from collagen by conversion into an unoriented protein by the method of partial hydrolysis [54]. According to the pH used during collage hydrolysis, gelation is classified into two types: gelatin A (isoelectric point of 9), where collagen is hydrolyzed at acidic pH, and gelatin B (isoelectric point of 5), where collagen is hydrolyzed at alkaline pH. Because of its water solubility, it may be necessary to cross-link gelatin during the development of nanoparticles [55,56]. They are characterized by controlling the release of loaded drugs [57].

#### 2.1.4. Albumin

Albumin is the most popular plasma protein. Its molecular weight is about 66,000 Da [51]. It is a biodegradable, nontoxic, and non-immunogenic protein. Therefore, it presents a potential carrier for drug delivery [49]. More precisely, it is considered a promising platform for cancer drug delivery as it can be preferentially entered into cancer cells (by endocytosis) to provide them with nitrogen and energy, allowing the loaded active to be delivered into cancers [58,59]. Nanoparticles prepared from albumin are generally stimuli-sensitive and highly bioactive substances [60]. 

### 2.2. Synthetic Polymers

In the literature, there are many synthetic polymers frequently utilized for the development of polymeric nanoparticles, such as polylactic acid, poly (lactide co-glycolides), polycaprolactone, and poly(amido amine) (Table 2). Different methods were used to prepare such types of nanoparticles (Figure 5).

#### 2.2.1. Polylactic Acid (PLA)

PLA is an aliphatic polyester characterized by its good biodegradability and biocompatibility because it can degrade in situ through hydrolysis. During the hydrolysis process, water molecules break the ester bonds in the PLA backbone, leading to the formation of lactic acid and its short oligomers (Figure 4). These degradation byproducts are easily metabolized by the body, offering an inherent biocompatibility that diminishes the accomplishment of immune reactions. Additionally, it can simply be processed with typical development methodologies, such as injection and extrusion [89]. Furthermore, it displays a high mechanical strength that can be effectively used to develop a controlled-release drug delivery system [90]. Nanoparticle-based PLA can be prepared by different methods such as emulsion-based, nanoprecipitation, spray drying, and supercritical fluid methods.

#### 2.2.2. Poly(lactic-co-glycolic Acid) (PLGA)

PLGA is considered one of the most effectively used biodegradable nanocarriers for the production of nanoparticles as it degrades inside the body by hydrolysis to yield lactic acid and glycolic acid, which are biodegradable metabolic products (Figure 4). As these two byproducts are endogenous and easily undergo biotransformation through the Krebs cycle, the toxicity accompanied by using PLGA for drug delivery is negligible [9,10]. It has recently been tested for use in cancer imaging and therapy [10]. However, the reassembly of PLGA into polymeric devices is still under consideration [91]. PLGA nanoparticles have generally been developed by solvent emulsion–evaporation [92], interfacial deposition [93], emulsification–diffusion [94], and the nanoprecipitation method [95].

#### 2.2.3. Poly-ε-caprolactone (PCL)

PCL is a semi-crystalline aliphatic polyester (Figure 4) that is widely used as a biocompatible and biocompatible platform for the development of nanoparticles. Owing to its hydrophobic nature, the encapsulation of hydrophobic actives is quite easy. Compared with other polymeric materials, the biodegradation of PCL is slower, giving a greater chance for developing controlled release delivery systems [96,97,98]. It can be used alone or as a blend with other biocompatible polymers to obtain new desired properties, such as degradation kinetics, hydrophilicity, and mucoadhesion [99,100]. For example, copolymerization with PLA and PLGA enhanced the biodegradation property [101]. PCL nanoparticles have been generally developed by several techniques, such as the emulsion solvent evaporation technique and the nanoprecipitation and dialysis methods.

#### 2.2.4. Poly(amidoamine) (PAMAM) 

This type can develop dendritic polymers possessing good biocompatibility, water solubility, and non-immunogenicity [102]. It can be loaded with different types of drug molecules owing to its distinctive architecture (amide and amide functionality containing polymer). Unfortunately, PAMAM dendrimers were reported to have certain toxicity at cellular levels [103], especially uncoated ones [104]. Thus, surface modification becomes very important to improve their safety profile. The PEGylation of PAMAM has been considered one of the most efficient methods of drug delivery for the treatment of cancer [105]. For example, the PEGylation of PAMAM dendrimers enhanced efficacy and mitigated the toxicity of anticancer and gene delivery [103]. Unlike other polymers, PAMAM dendrimers are synthesized by stepwise synthesis procedures (such as using divergent or convergent methods or a combination of both methods [106,107,108]), which donate well-arranged structures and narrow polydispersity.

## 3. Why Do We Need Polymeric Nanoparticles for Delivery of Natural Bioactive Agents? 

Natural bioactive agents generally face problems such as poor aqueous solubility, poor stability in alimentary canal conditions, extensive metabolism, short half-life, and low bioavailability [109]. The chemical structure of the natural bioactive agents regulates their absorption after oral administration and plasma concentrations. However, the determination of plasma concentration is a key indicator for establishing the bioavailability of natural bioactive agents and their metabolites. For example, nearly 10% of orally administered polyphenols, or their metabolites, are detected in urine and plasma [110]. Other problems associated with oral delivery are insufficient gastric transit period, poor intestinal permeability, and sensitivity to alimentary canal conditions (pH, enzymes, etc.), all of which obstruct the efficacy and therapeutic effectiveness of these natural bioactive agents [111]. If the natural bioactive agent is suffering from extensive metabolism, oral administration will result in a faint therapeutic effect due to the poor availability of active form in the plasma or the availability of pharmacologically inactive fractions. 

Establishing a suitable drug delivery system is a major challenge for loading natural bioactive agents. Generally, compounding in nanocarriers is a promising strategy for the delivery of such compounds because the purpose of the nanosized delivery systems of natural bioactive agents is to improve their efficacy and overwhelm the drawbacks associated with their administration. In particular, polymeric nanoparticles are characterized by several properties that enable them to defeat most of the challenges facing the delivery of natural bioactive agents by different routes. For instance, for oral delivery, they can protect actives from degradation and convey them to the optimal regions within the GIT [112]. In addition, their small particle size results in possessing a large effective surface area, leading to an improvement in the dissolution rate of loaded drugs and the effective interaction with epithelial surfaces [9,113]. Furthermore, adjusting self-assembly conditions during preparing polymeric nanoparticles and selecting particular polymer categories can lead to an appropriate design by controlling their characteristics (such as particle diameter, zeta potential, and hydrophobicity) and drug release parameters (fast, controlled, sustained) [114]. In the case of parenteral use, surface modification of polymeric nanoparticles is feasible and not complicated, which leads to overcoming some obstacles associated with parenteral administration. In particular, surface decoration with targeting moieties and PEGylation can effectively lead to targeting behavior and longer residence time in general circulation, respectively. This can be achieved by using diverse polymers tailored with desired groups or coupling specific polymers to the designed polymeric nanoparticles [115]. Specifically, polymeric nanoparticles were reported to effectively deliver actives to the brain through susceptibility to adsorptive-facilitated transcytosis and receptor-mediated transcytosis across the blood–brain barrier, which can be achieved by targeting peptides and/or cell-penetrating moieties anchored to surfaces of polymeric nanoparticles [116]. Different mechanisms were also reported by Kreuter [117,118], such as endothelial cell transcytosis, endothelial cell endocytosis, high-concentration gradients created by polymeric nanoparticles, and membrane fluidization by included surfactants. In the case of topical delivery, polymeric nanoparticles improve skin permeation as their assembly on the surface of the skin or hair follicle can offer a reservoir to release the drug incessantly, creating a concentration gradient [119,120].

## 4. Natural Bioactive Agents Loaded Polymeric Nanoparticles

Various natural bioactive agents have been loaded into polymeric nanoparticles intended for different uses, as mentioned below. 

Bioflavonoids are one of the most popular classes of natural bioactive agents that gained attention due to their promising therapeutic activities and health-promoting effects [121]. The health-improving effect enhances well-being and decreases the opportunity of emerging risk, which includes the immune-stimulant effect, cytotoxic effect, valuable cardiovascular influence, improved eyesight [122], and reduction in amyloidogenic-related cell death [123]. In addition, they can provide efficient anticancer activity by different mechanisms (Figure 6). Their actions were utilized to treat different diseases such as oxidative stress, inflammation, skin disorders, cardiovascular, respiratory, and diabetes.

### 4.1. Quercetin

Quercetin (3,3′,4′,5,7-penthydroxyflavone) is semi-lipophilic flavonol plentifully present in leafy vegetables, citrus fruits, tomatoes, berries, etc. It possesses several therapeutic effects such as antioxidant, anti-inflammatory, wound healing, antithrombotic, antitumor, antiprotozoal [124,125,126,127,128], and cytoprotective effects [129]. Among the different nanocarriers, polymeric nanoparticles may provide an effective way to encapsulate quercetin as they can improve bioavailability, therapeutic effectiveness, and in vivo stability. However, encapsulation of quercetin in polymeric nanoparticles was reported not to influence their efficacy. Kumari and coworkers developed quercetin-loaded PLA nanoparticles and reported a similar antioxidant activity to free quercetin [67]. Sunoqrot and Abujamous developed the pH-sensitive polymer Eudragit S100 loaded with quercetin by a nanoprecipitation technique to achieve potential colon-specific drug delivery [130]. The prepared formula was 66.8 nm in particle size, −5.2 mV zeta potential, and 41.8% encapsulation efficiency. It showed minimal in vitro quercetin release at an acidic pH and complete release at pH 7.2. In vitro cytotoxicity assay showed promising activity against CT26 murine colon carcinoma cells and significantly lower IC50 when compared to free quercetin. Zhou and coworkers quercetin loaded PLGA-TPGS nanoparticles to improve anticancer activity [131]. The prepared formula was 198.4 ± 7.8 nm particle size, −22.5 ± 2.5 mV zeta potential, and 82.3 ± 5.7% encapsulation efficiency. In vitro cytotoxicity assay exhibited a potential activity against triple-negative breast cancer cells. Interestingly, after oral administration, significant anticancer activity was found in 4T1-bearing mice with few metastatic lung colonies. Additionally, the inhibitory activity on the migration of urokinase-type plasminogen activator (breast cancer tumor marker) knockdown MDA-MB231 cells was significantly decreased. Elmowafy and coworkers prepared and optimized quercetin-loaded mesoporous silica nanoparticles in order to improve the physiochemical characteristics, particularly the dissolution rate and hence the bioavailability [132]. The optimized formulation (using 3^2^ full factorial design) exhibited good physiochemical properties and enhanced the saturation solubility and dissolution rate. Pandey and coworkers developed quercetin-loaded PLA nanoparticles to improve anticancer activity against breast cancer [66]. The particle sizes of the prepared formulations varied from 32 ± 8 to 152 ± 9 nm with 62 ± 3% (*w*/*w*) encapsulation efficiency. In vitro cytotoxicity experiment showed that 5 days were sufficient to kill about 40% of breast cancer cells, verifying a sustained release manner. Zhang and coworkers developed quercetin-loaded chitosan nanoparticles to improve antioxidant activity and bioavailability [133]. They found that quercetin existed in nanoparticles in an amorphous state, which showed good antioxidant activity evaluated by scavenging DPPH radical and reducing the power of some plant extracts. 

### 4.2. Curcumin

Curcumin is a natural hydrophobic yellow polyphenol derived from the turmeric rhizome. Accumulating studies have revealed that curcumin possesses a wide range of therapeutic activities. In the previous century, studies proved different pharmacological actions, including antihypercholesterolemia [134], antidiabetic [135], anti-inflammatory [136], and anti-oxidant [137] actions. Recently, different beneficial actions have been reported, such as metabolic disorders [138], inflammatory bowel diseases [139], anti-hepatotoxic [140], anti-proliferative [141,142], anti-carcinogenic [142], antimicrobial [143], and neuroprotective properties [144]. While it looks to have numerous therapeutic activities, its practical applications have been limited owing to limited aqueous solubility, poor bioavailability, rapid biotransformation, low penetrative and targeting power, and instability in alkaline conditions and in the presence of metal ions, heat, and light [145]. In order to improve the oral bioavailability of curcumin, Umerska and coworkers compared three polymeric nanoparticles, Eudragit^®^ RLPO, PLGA, and PCL, loaded with curcumin [146]. The authors found that Eudragit^®^ RLPO nanoparticles had smaller particle sizes, lower encapsulation efficiency, and faster release within 1 h. All developed formulations exhibited good compatibility with intestinal Caco-2 cells. They also concluded that PLGA and PCL-based nanoparticles are more suitable for parenteral administration. Jahagirdar and coworkers developed PLGA nanoparticles loaded with curcumin by facile in situ nanoprecipitation method to target macrophages and treat associated diseases such as intracellular infections [147]. The optimized formulation was 208.25 ± 7.55 nm in particle size, −26  ±  2.55 mV zeta potential, and 90.16 ± 1.17% encapsulation efficiency. Structural analyses (by DSC and FTIR) showed that curcumin was embedded in polymeric matrix in amorphous state. Flow cytometry and HPLC analyses verified fast and efficient uptake (three-fold enhancement in uptake when compared to free curcumin) by RAW 264.7 macrophages. Confocal microscopy confirmed localization of nanprticles in phago-lysosomal compartment. Udompornmongkol and Chiang developed curcumin loaded chitosan/gum arabic polymeric nanoparticles by an emulsification solvent diffusion technique to improve anti-colorectal cancer activity [148]. The developed formulation was 136.3 ± 3.9 nm in particle size, +48 mV zeta potential, 3.57 ± 0.69 loading efficiency, and 950.2 ± 0.03% encapsulation efficiency. Significant cell viability reduction against two models of human colorectal carcinoma cells (HCT116 (Duke’s type A)) and HT29 (Duke’s type B)) and cellular uptake (by fluorescence microscopy and flow cytometry). In addition, curcumin polymeric nanoparticles were more proficient in inducing apoptosis when compared to free curcumin. Nair and coworkers developed curcumin-loaded chitosan polymeric nanoparticles by ionic gelation (using tripolyphosphate as a cross-linker) technique to improve transdermal delivery [149]. The authors studied different factors (such as the chitosan/tripolyphosphate ratio and stirring rate) on the physicochemical characteristics of the developed batches. They found that a high chitosan/tripolyphosphate ratio produces larger particles as tripolyphosphate aided in the attraction of chitosan molecules by sturdier intramolecular interactions, leading to the formation of reduced particle sizes. A high stirring rate (1000 rpm) resulted in larger particle sizes due to the decrease in the repulsive forces between the particles and the improved aggregation. Additionally, curcumin polymeric nanoparticles showed enhanced permeation through Strat-M^®^ membrane and percentage cell viability towards human keratinocyte (HaCat) when compared to free curcumin solution [149]. Recently, Walbi and coworkers developed and optimized (using a 3-factorial Box–Behnken statistical design) curcumin-loaded lecithin/chitosan polymeric nanoparticles to improve drug loading and controlled release behavior [150]. The optimized formulation was of 138.4 ± 2.10 nm particle size, +18.98 ± 0.72 mV zeta potential, and 77.40 ± 1.70% encapsulation efficiency. It also showed diffusion-controlled release behavior and good stability.

On the other hand, researchers paid attention to the functionalization of polymeric nanoparticles’ surfaces in order to improve efficacy and biocompatibility. Grabovac and Bernkop [151] synthesized thiolated chitosan tailoring curcumin PLGA nanoparticles. Thamake and coworkers [152] synthesized homobifunctional spacer, bis(sulfosuccinimidyl) suberate (BS3) tailoring curcumin PLGA nanoparticles and reported an enhancement linkage of Annexin A2 leading to proficient specific delivery to Annexin-A2-positive MDA-MB-231 cancer cells. Joseph and coworkers developed PEG-modified PLGA nanoparticles loaded with curcumin by nanoprecipitation technique to target the brain and control hypoxic–ischemic brain injury in neonatal rats using the Vannucci model [153]. Results showed that nanoparticles could overcome the impaired blood–brain barrier and improve the extravasation of nanoparticles into the brain parenchyma, leading to an efficient decrease in global injury by the neuroprotective effect, which was more marked in the penumbral region. Anitha and coworkers developed curcumin-loaded water-soluble chitosan derivatives (O-carboxymethyl chitosan and N,O-carboxymethyl chitosan) and showed good biocompatibility and induced apoptotic cancer cell death [154,155]. The development of multiple polymeric materials nanocomposites was also a point of interest. Das and coworkers prepared curcumin-loaded alginate/chitosan/Pluronic nanocomposite by ionotropic pre-gelation followed by polycationic crosslinking, showing cellular internalization inside the Hela cells (verified by fluorescent microscopy) [156]. A combination of natural and synthetic polymers in a single polymeric matrix was also studied by Liu and coworkers. They developed chitosan/PCL polymeric nanoparticles by the nanoprecipitation method, showed positively charged nanoparticles, and enhanced cellular uptake by Hela cells and human choroidal melanoma. The authors proposed that the combination of PCL with chitosan (bioadhesive polymer) led to efficient curcumin loading and release profiles, cellular interaction, and remarkable in vitro anticancer activity [157].

### 4.3. Kaempferol

Kaempferol (3,5,7-trihydroxy-2-(4-hydroxyphenyl)-4H-1-benzopyran-4-one) is frequently present in different plats such as onions, green tea, and grapes. Kaempferol has promising pharmacological actions such as antitumor, antioxidant, anti-inflammatory, and anti-allergic effects [158,159,160]. Like other flavonoids, its therapeutic applications are limited because of its limited aqueous solubility, poor oral bioavailability, extensive first-pass metabolism, and instability in water alkaline medium [161].

Luo and coworkers developed Kaempferol-loaded nonionic poly(ethylene oxide)-poly(propylene oxide)-poly(ethylene oxide) (PEO-PPO-PEO) and PLGA nanoparticles as chemopreventive of ovarian cancer [162]. The authors concluded the selectivity of PLGA nanoparticles to cancer cells was not offered by PEO-PPO-PEO nanoparticles (both cancerous and normal cells were affected). Kazmi and coworkers developed Kaempferol-loaded hydroxypropyl methylcellulose acetate succinate and Kollicoat MAE 30 DP nanoparticles to control induced hepatocellular carcinoma [163]. The authors prepared several batches and obtained a nanometric range of particle size, monodisperse particles, and negatively charged nanoparticles. They also found a significant decrease in serum levels of different liver biomarkers and an increase in antioxidant enzymes and proteins. Additionally, significant decreases in an mRNA expression of interleukin-1β, interleukin-6, and tumor necrosis factor-alpha verify a potential application against oxidative stress induced-liver injury. Ilk and coworkers developed Kaempferol loaded chitosan nanoparticles to modulate quorum sensing mediated by auto-inducers in model bioassay test systems to be used as new emerging strategy in antimicrobial chemotherapy [164]. Nanoparticles were developed by ionic gelation and were 192.27 ± 13.6 nm and +35 mV in value for particle size and zeta potential, respectively. Nanoparticles significantly prohibited the production of violacein pigment *in Chromobacterium violaceum* CV026 within the 30 storage days. In another work, Ilk and coworkers developed Kaempferol-loaded lecithin/chitosan nanoparticles by electrostatic self-assembly technique to attain sustained antifungal activity [165]. Nanoparticles significantly prohibited 67% of the growth of *Fusarium oxysporium* within the 60 storage days.

### 4.4. Resveratrol

Resveratrol (3,5,4′-trihydroxystilbene) is a natural non-flavonoid polyphenolic substance found in different plants, such as grapes, peanuts, and legumes [166]. It exists in two isomers: cis-(Z) and trans-(E) isomers, where the trans-isomer is considered a more stable form and biologically active than the cis-isomer because of its non-planar conformation [167,168]. However, the trans-isomer can be converted to the cis-isomer form upon exposure to UV-light [169,170]. Resveratrol exhibits several biological actions, such as cardioprotection, cancer prevention, platelet de-aggregation, antioxidant, anti-inflammatory, and vasorelaxant properties [171]. It can also improve the antiviral activity of zidovudine, zalcitabine, and didanosine [172]. 

Hung and coworkers developed PEG-modified PLA nanoparticles loaded with resveratrol to enhance drug stability, controlled delivery, and anticancer activity [173]. Cellular apoptosis and uptake were enhanced in nanoparticles when compared to free resveratrol (verified by flow cytometry and western blots). In vivo, retardation of tumor growth was observed during a long treatment period. Yee and coworkers developed methoxyPEG-modified PLA nanoparticles loaded with resveratrol to improve its plasmatic stability and reduce the metabolic rate [61].

The developed nanoparticles displayed good stability in plasma and decreased hepatic metabolic rate. The authors reported a correlation between enhanced plasma stability, reduced liver metabolic rate, and suppression of tumor growth in mice. Chen and coworkers developed PLGA nanoparticles loaded with both resveratrol and puerarin by the O/W emulsion method to improve their chemotherapeutic efficacy in spinal cord injury [174]. Nanoparticles were homogeneously dispersed with mean diameters varying from 238 nm to 274 nm and a zeta potential of −12.6  ±  2.1 mV. Optimized formulation showed 72−79% of drug release through 36 h and decreased free radicals production and oxidative stress. Zu and coworkers developed carboxymethyl chitosan nanoparticles loaded with resveratrol by emulsion cross-linking to improve its physicochemical properties, antioxidant activity, and pharmacokinetic profile [175]. Optimized formulation was of 155.3 ± 15.2 nm, −10.28 ± 6.4, and 44.5 ± 2.2% values for particle size, zeta potential, and encapsulation efficiency, respectively. It also showed improved solubility and antioxidant activity. In addition, it showed enhanced in vivo absorption, prolonged duration of action, and increased relative bioavailability by 3.5-fold when compared to free drugs. Sarma and coworkers developed chitosan–pectin core–shell nanoparticles loaded with resveratrol to improve its antioxidant activity and obtain prolonged release [176]. Liu and coworkers developed α-lactalbumin and chitosan core–shell nanoparticles loaded with resveratrol [177]. Nanoparticles were more stable against UV-light and heat and bioaccessible with prompt antioxidant activity.

### 4.5. Epigallocatechin-3-Gallate

Epigallocatechin-3-Gallate (a major constituent of green tea polyphenols) has been shown to be chemopreventive for various kinds of cancers, such as liver, prostate, stomach, and breast cancers [178,179,180,181]. 

Sanna and coworkers developed PLGA and PLGA–PEG nanoparticles loaded with (-)-epigallocatechin-3-Gallate for targeting prostatic cancer by nanoprecipitation method [182]. All prepared batches were homogeneously dispersed with mean diameters varying from 130 nm to 250 nm and negatively charged surface charges (ranging from −32.1  ±  8.80 to −21.3  ±  8.35 mV). In vitro, nanoparticles showed an enhanced anti-proliferative activity against PCa cell lines and apoptosis modulation when compared to the free drug. In vivo, functionalization of PLGA nanoparticles with low molecular weight molecules led to specific targeting of prostate-specific membrane antigen and improved anticancer activity in the mouse xenograft model of the prostatic tumor. 

Alserihi and coworkers developed folic acid anchored PLGA–PEG nanoparticles loaded with (-)-epigallocatechin-3-Gallate by nanoprecipitation for prostatic cancer treatment [183]. The prepared nanoparticles were 128.42 ± 3.57 nm and 0.114 ± 0.03 and 49.82 ± 3.74% in particle size, polydispersity index, and encapsulation efficiency, respectively. Results also exhibited a significant enhancement in the antiproliferative activity against prostate cancer cell lines, particularly toward the prostate-specific membrane antigen, and an increase in the number of dead apoptotic cells in 22Rv1 spheroids. Shao and coworkers developed nanofiber meshes by electrospinning the PCL loaded with green tea polyphenol to increase the efficacy against different cancerous lines [184]. The prepared nanoparticles showed a controlled release pattern and longer stability of green tea polyphenol. It also showed significant inhibitory effects against HepG2 and A549. 

Young-Jin and coworkers developed PCL nanofibers loaded with (-)-epigallocatechin-3-gallate and caffeic acid to increase the efficacy against gastric cancer cell lines (MKN-28). The prepared nanoparticles showed apoptosis via activated caspase-3 due to the H_2_O_2_ generation, which offered a suitable system for long-term cancer therapy [185]. Safer and coworkers fabricated chitosan nanoparticles loaded with (-)-epigallocatechin-3-Gallate by ionic gelation method in order to develop a nano-therapeutic delivery system against hepatic fibrosis [186]. The prepared nanoparticles were stable, positively charged, in the nanorange, and had isoelectric pH of 7.61. Moreno-Vásquez and coworkers fabricated (-)-epigallocatechin-3-gallate grafted-chitosan nanoparticles by nanoprecipitation method in order to maximize antibacterial and antioxidant potential [187]. FTIR finding verified the modification of the chitosan with (-)-epigallocatechin-3-gallate where nanoparticles fabricated offered lower minimal inhibitory concentration when compared to chitosan and (-)-epigallocatechin-3-gallate against Pseudomonas fluorescens. In addition, nanoparticles produced a significant enhancement of reactive oxygen species (*p* < 0.05) in two bacterial species and an increase in antioxidant activity. Hong and coworkers developed (-)-epigallocatechin-3-gallate-loaded self-assembled nanoparticles of chitosan and aspartic acid in order to improve its action against atherosclerosis [188], which was found to be comparable with simvastatin administration in the percentage of lipid deposition. Liang and coworkers developed (-)-epigallocatechin-3-gallate-loaded chitosan/β-Lactoglobulin nanoparticles of chitosan and aspartic acid in order to achieve a prolonged release of epigallocatechin-3-gallate in the gastrointestinal tract [189]. 

### 4.6. Silymarin

Silymarin is an active extract from milk thistle seeds that consist of 65–80% silymarin flavonolignans. It has been used as a natural medication to protect the liver and treat several liver diseases, such as hepatitis and cirrhosis. Pharmacologically, it has antioxidant, antilipid peroxidative, antifibrotic, and anti-inflammatory actions. Additionally, it has been widely studied as a chemopreventive agent against different cancers [190,191,192]. Snima and coworkers fabricated PLGA nanoparticles loaded with silymarin by single-step emulsion technique to improve the anticancer application in vitro. Nearly 60% of silymarin was entrapped into PLGA nanoparticles, which produced time- and dose-dependent cytotoxic effects against prostate cancer cells (PC-3) [193]. Azadpour and coworkers fabricated PLGA nanoparticles loaded with silymarin by the emulsification/solvent evaporation method to improve the anti-inflammatory effect against sepsis and septic shock [194]. Results showed that nanoparticles decreased pro-inflammatory cytokines and increased anti-inflammatory cytokine and M2-associated markers and proteins. Elsherbiny and coworkers fabricated pH-responsive alginate–PLGA nano/micro hydrogel nanoparticles loaded with silymarin to develop a biodegradable and controlled-release oral drug delivery system for silymarin [195]. Results showed that the developed alginate-based hydrogel microparticles encapsulating silymarin-loaded PLGA nanoparticles can improve the dissolution and oral bioavailability of silymarin.

### 4.7. Saponins

Saponins are a group of naturally derived plant glycosides that demonstrated considerable cytotoxic activity [196,197,198]. According to their origins, there are different types, such as *Astragalus saponin, Ginseng saponin, and* saponin extracts from the *Quillaja saponaria Molina.* Sanoj and coworkers developed saponin-loaded chitosan nanoparticles by ionic gelation as a biocompatible and biodegradable system to control and sustain delivery to prostate cancer cell lines. The cellular internalization studies showed a promising uptake by L929 and PC3 cell lines with potential selective cytotoxic activity against cancerous [26]. Van de Ven and coworkers developed saponin-loaded chitosan nanoparticles in order to improve its release and anticancer activity [26]. Results showed that nanoparticles have a particle size of 65 ± 7 nm with improved cytotoxicity on PC3 and KB cell lines and cellular uptake by L929 and PC3. Saharan and coworkers developed saponin β-aescin loaded PLGA nanoparticles in order to improve drug delivery in Leishmania-infected macrophages [199]. Nanoparticles were prepared by combined emulsification solvent evaporation/salting-out technique. Results showed that nanoparticles have a particle size of less than 300 nm with negative zeta potential. Additionally, they were up-taken by the macrophages and trafficked towards the lysosomes. 

### 4.8. Oridonin

Oridonin is a natural diterpenoid that was reported to have several potential therapeutic actions against cancerous cells, including induction of apoptosis in lymphoid malignancies, inhibition of Nuclear factor-kB (NF-kB ), and down-regulation of the Bcl-2 family proteins [200]. Unfortunately, the clinical applications of oridonin are really limited owing to its poor aqueous solubility and narrow therapeutic index. Xing and coworkers developed oridonin-loaded PLA nanoparticles by an emulsion solvent diffusion method in order to improve its plasma residence period and drug delivery [201]. The prepared nanoparticles showed a good pharmacokinetic profile; particularly in the long period, the drug stayed in circulation. It also showed significant deposition of oridonin in spleen, liver and lung, while its renal and cardiac concentrations were noticeably dropped. 

### 4.9. Paclitaxel

Paclitaxel is a natural product isolated from the bark of Pacific Yew (*Taxus brevifolia*). It is widely used as a chemotherapeutic agent in the treatment of different types of cancers, such as advanced ovarian, lung, and breast cancers [202,203]. The mechanism by which paclitaxel acts is the stabilization of microtubules, blocking the cancer cell cycle in the G2/M phase, and finally, the induction of apoptosis [204]. The major obstacle to paclitaxel delivery is its poor aqueous water solubility (~0.4 μg/mL). However, this drawback was overcome in the commercial product (Taxol^®^) by dissolving paclitaxel in Cremophor EL/ethanol (1/1, *v*/*v*). Nevertheless, Cremophor EL was reported to have various severe adverse reactions, including hypersensitivity, cardiotoxicity, and peripheral neuropathy [205]. 

Danhier and coworkers developed paclitaxel-loaded PEGylated PLGA nanoparticles by nanoprecipitation in order to develop Cremophor EL-free i.v. formulation to enhance the safety and therapeutic index of paclitaxel [206]. The prepared nanoparticles showed biphasic release pattern, significant in vitro anticancer efficacy against human cervix carcinoma cells (HeLa), and lower IC50 (5.5 µg/mL) when compared to commercial product (15.5 µg/mL), concentration-dependent cellular uptake and comparable apoptosis. In vivo, nanoparticles presented superior tumor growth inhibition outcomes on TLT tumors.

Fonseca and coworkers [207] developed paclitaxel-loaded PLGA nanoparticles by the interfacial deposition method in order to achieve the same target as the previous study. Results showed that the prepared nanoparticles were less than 200 nm in particle size, mono-dispersed, and carried a negative charge. The prepared nanoparticles showed a biphasic release pattern and significant in vitro anticancer efficacy against small cell lung cancer cell line (NCI-H69 SCLC). Su and coworkers [208] developed paclitaxel-loaded cholic acid-functionalized star-shaped PLGA-d-α-tocopheryl polyethylene glycol 1000 succinate nanoparticles by modified nanoprecipitation method in order to treat malignant melanoma. Authors reported that nanoparticle formulation overweighed commercial products in treating malignant melanoma in terms of in vitro cell toxicity and xenograft tumor model in vivo. Cui and coworkers [209] developed magnetic PLGA nanoparticles co-loaded with paclitaxel and curcumin by modified nanoprecipitation method in order to cross the blood–brain barrier effectively and treat glioma. A dual-targeting approach was utilized by a combination of magnetic guidance and active targeting transferrin receptor-binding peptide T7-mediated nanoparticles. The resultant nanoparticles resulted in a more than 10-time increase in cellular uptake studies, a more than 5-time enhancement in brain delivery compared to the non-targeting nanoparticles and high survival rate in mice bearing orthotopic glioma. 

Abriata and coworkers [210] developed PCL nanoparticles loaded with paclitaxel by nanoprecipitation method for evaluation of the activity against ovarian cancer in vitro. The results showed that nanoparticles had a mono-modal particle distribution with a size of around 140 nm, negatively charged particles, and high entrapment efficiency. They also presented a high efficiency in reducing the cell viability of SKOV-3 cells. Bernabeu and coworkers [211] developed PCL-alpha tocopheryl polyethylene glycol 1000 succinate (TPGS) nanoparticles loaded with paclitaxel to fabricate long-circulating safe nanoparticles and evaluate the activity against breast cancer in vitro. The authors prepared nanoparticles by three different methods (nanoprecipitation, emulsion–solvent evaporation/homogenization, and emulsion–solvent evaporation/homogenization). The former method yielded smaller particles and higher encapsulation of paclitaxel when compared to other methods. The results presented a better efficiency in reducing cell viability of two human breast cancer cell lines (MCF-7 and MDA-MB-231) when compared to free paclitaxel solution and the commercial product Abraxane^®^). Gupta and coworkers [212] developed chitosan nanoparticles loaded with paclitaxel to achieve safe and effective delivery. The authors prepared nanoparticles by w/o nanoemulsion method, and the yielded formulation was of 226.7 ± 0.70 nm particle size, 0.345 ± 0.039 polydispersity index, 37.4 ± 0.77 mV surface charge, 79.24 ± 2.95% encapsulation efficiency, and 11.57 ± 0.81% loading capacity. The results showed promising efficiency in reducing cell viability of human breast cancer cell line MDA-MB-231, a reduced IC50, and extreme biocompatibility. 

## 5. Polymer Functionalization and Surface Decoration

The functionalization of polymeric nanoparticles has garnered much attention in previous years through different methods of polymeric modification. The surface decoration, chemical synthesis, and the selection of the nanoparticles are completely dependent on the nature of the actives, the required duration of action, stability, permeability, the release of the actives, etc. [213]. However, the aims of polymeric functionalization and surface decoration depend on the objective of the work. It was reported that the functionalized polymeric nanoparticles are mostly classified into four generations; long circulating “stealth” nanoparticles, lectin-based polymeric nanoparticles, polysaccharide-based polymeric nanoparticles, and, lately, ligand-based polymeric nanoparticles [214]. Generally, they aim to prolong the release and plasma residence time, specifically deliver the actives by active targeting, enable nanoparticles to withstand harsh environments, and improve physicochemical properties and biocompatibility. On the other hand, researchers should take the following criteria into consideration during working on the functionalization of polymeric nanoparticles [215]: Determination of the number of conjugate sites to calculate the bio-molecule ratio appropriately.Avoidance of non-specific conjugation.Adjustment of polymer/conjugate affinity.Keeping the proper effectiveness of the physiochemical properties of the yielded polymer/conjugate.High reproducibility of the method.

There are two main techniques used for the functionalization of the polymers: direct functionalization and post-functionalization. In the first technique, a reactive group is attached to the nanoparticle surface from one side and to the required active group from the other side. Frequently, polymers have been functionalized with different groups, such as thiols, disulfides, and amines [216,217,218]. 

Coester and coworkers fabricated biofunctionalized gelatin nanoparticles with sulfhydryl groups to improve the entrapment of biotinylated substances because the sulfhydryl group was covalently linked with proteins such as avidin [219]. Lectins are essential to and easily recognized by several cells. Therefore, they display selective non-covalent interaction with carbohydrates. However, they can be incorporated into polymeric nanoparticles by adsorption or covalent linkage. Rodrigues and coworkers fabricated biofunctionalized PCL-based nanoparticles as lectin-decorated or protein-loaded nanoparticles with PCL as a hydrophobic core and dextran as a hydrophilic corona [220]. 

Lectins were bovine serum albumin, and those obtained from Bauhinia monandra, Lens culinaris leaves, and the prepared nanoparticles were of nearly 200 nm particle size. They also conserved lectins’ hemagglutinating activity and enhanced Caco-2-cellular viability, proposing the likely use of this modified nanoparticle type for targeted oral delivery. In the second method, a similar strategy is followed, and the nature of the functionalizing moiety may not be compatible with good control over the size and dispersion in the vehicle used during the preparation [215]. As most biodegradable polymers are hydrophobic in nature, they are easily recognized by the mononuclear phagocytic system to remove them from the blood stream before reaching the target. However, the fabrication of nanoparticles of particle size less than 100 nm was reported to escape from mononuclear phagocytic system recognition [221]. This strategy is not usually guaranteed for most polymeric nanoparticles. Thus, surface coating with a hydrophilic polymer (such as polyethylene glycol) will offer the chance for nanoparticles to escape from mononuclear phagocytic system recognition by forming a hydrophilic shielding layer surrounding the nanoparticles, which enables deterring the attraction of opsonin proteins by steric repulsive forces, thus closing and suspending the first stage in the opsonization process [222]. In addition, and depending on the anchored polymer, surface modification can improve the targeting of tumors and resistance to GIT-harsh conditions. Hydrophilic polymers such as polyethylene glycol, Pluronics, Tween 80, TPGS, and Tween 20, and polysaccharides such as dextran are frequently used to modify the surface of conventional nanoparticles [223]. 

Surface coverage by hydrophilic polymers can be achieved by surface adsorption or by the use of block or branched copolymers [224,225]. Polyethylene glycol is a hydrophilic polymer and is commonly used for coating the surface of different polymeric nanoparticles [226] due to its good biocompatibility. It can also prolong the residence time of nanoparticles in plasma efficiently due to escaping from mononuclear phagocytic system recognition. Kim and coworkers fabricated methoxy polyethylene glycol surface decorated PCL nanoparticles [227]. Calvo and coworkers fabricated polyethylene glycol surface decorated polyhexadecyl cyanoacrylate nanoparticles to minimize the opsonization and recognition by macrophages and, thus, prolong the plasma half-life. PEGylated nanoparticles presented reasonably higher deposition in the spleen and brain than conventional non-PEGylated nanoparticles [228]. Tobio and coworkers fabricated a polyethylene glycol surface decorated PLA nanoparticles for oral administration to improve the stability and decrease the influence of GIT on the digestive enzymes [226]. However, the selection of the more relevant PEG chain is not an easy process as various brands of PEG chains possess dissimilar physicochemical properties (such as molecular weight, chain length, and several branch arms), which could influence the efficiency of the PEG chain. It was reported that PEG with a molecular weight of less than 2000 Da was not efficient enough to prevent protein adsorption and immuno-recognition [222,229,230]. On the other hand, no extra benefit was achieved upon using a PEG chain with a molecular weight higher than 5000 Da [231]. Pluronics (Poloxamers) were reported to improve plasma residence time and deposition in tumors of polyethylene oxide–PCL nanoparticles [232]. Tweens-coated nanoparticles were reported to adsorb certain compounds from the blood by endothelial cells facilitating the transport through the blood–brain barrier [233,234,235]. Dextran was also reported to prohibit protein adsorption of PCL nanoparticles [236]. TPGS was reported to provide milder environments to proteins when compared to uncoated PLA nanoparticles [237].

## 6. Clinical Trials

As polymeric nanoparticles have garnered much attention as an efficient drug delivery system, clinical trials using polymeric nanoparticles loaded with natural bioactive agents have been conducted. In addition, some formulations have been approved by Food and Drug administration (FDA). Paclitaxel albumin-stabilized nanoparticle has completed Phase II clinical trials for the treatment of ovarian cancer (NCT00499252 at https://clinicaltrials.gov/ and updated on 11 January 2018). On the other hand, two commercial products are now available in the market. Genexol PM^®^ are paclitaxel-loaded PEG-poly(D,L-lactide) nanoparticles at a dose of 30 mg and have been approved in South Korea (2007) for the treatment of breast cancer, though it is still under phase II clinical trials in the USA [238]. Paclical^®^ are paclitaxel-loaded poly (glutamic acid) nanoparticles for the treatment of ovarian cancer [239].

## 7. Challenges

Generally, using nanoparticles for drug delivery might be considered a risk as the particle size reduction is correlated to improved reactivity and toxicity [240]. However, other different factors can participate in such behavior, including morphology, chemical composition, hydrophobicity/hydrophilicity, and the surface charge of nanoparticles. Gatoo and coworkers studied the link between the physicochemical characteristics of nanoparticles and their induced toxicity. They found that the smaller the size of the particles, the higher the risk of potential toxicity, owing to their potential capability to traverse biological membranes and attain various tissues without being recognized by the reticuloendothelial system [241]. In particular, the safety/toxicity of polymeric nanoparticles is considered one of the major challenges. The toxicity of polymeric nanoparticles is attributable to the toxic monomer aggregation, homocompatibility, residual material attached to them, and toxic degradation products [242].

However, the toxicity of polymeric nanoparticles is particularly influenced by the quantum size, which is associated with oxidative stress, cytotoxicity, and genotoxicity [243]. In addition, the polymeric material using physicochemical properties of nanoparticle biodegradability and biocompatibility are also causal factors. Generally, the toxicological effects associated with natural polymer-based nanoparticles were established to be less than that associated with the administration of synthetic polymer-based nanoparticles [244]. In addition, using natural polymer blends or coating synthetic polymeric nanoparticles with natural polymer was reported to alleviate the toxicity [245,246,247,248,249]. The particle size of nanoparticles can also influence the manner of endocytosis, uptake, and efficacy of treatment in the endocytic pathway [250,251]. Commonly, as the size decreases, the opportunity for interaction with the biological membranes increases [252,253]. It was established that when particle size is bigger than 100 nm, particles can enter the cells via phagocytosis, whereas when the particle size is less than 100 nm, endocytosis is the main route of entry, leading to significant higher toxicity risk [254,255]. On the other hand, the incorporation of stabilizers (such as polyvinyl alcohol and Pluronic) was documented to exaggerate the toxicity of PLGA-based nanoparticles especially when used in high concentrations [256]. The surface charge might also have an impact in that regard. Regardless of their compositions, polymeric nanoparticles carrying positive charges are subjected to higher phagocytic uptake than negatively charged ones resulting in higher toxicological aspects [257]. Das and coworkers studied the differential pulmonary toxicity of positively and negatively charged PEG–PLA nanoparticles in vivo. Positively charged nanoparticles were reported to induce higher local and systemic toxicities [258]. Bancos and coworkers studied the effect of silica nanoparticles on macrophages and found that phagocytosis decreased and cytokine production increased [259]. Gonzalez and coworkers found that silica nanoparticles had cytotoxic effects on lung carcinoma cells [260]. Genotoxic effects on lung cells were associated with silica nanoparticle administration, which was reported by Maser and coworkers in a dose-dependent manner [261]. The production of reactive oxygen species was also one of the toxicological features of polymeric nanoparticles. Production of reactive oxygen species was reported in a dose-dependent manner upon administration of PLGA nanoparticles [256,262]. On the contrary, Patel and coworkers found no production of reactive oxygen species upon administration of PLGA nanoparticles [263]. Xifei and coworkers studied the effect of silica nanoparticles on oxidative stress and found the overproduction of reactive oxygen species with the exhaustion of glutathione at intracellular levels [264]. Immune reactions and the overproduction of inflammatory mediators are considered one of the most common toxicological aspects associated with polymeric nanoparticle administration. PLGA-polyethylene oxide nanoparticles were reported to increase GM–CSF levels (pro-inflammatory mediator) [265].

In addition, using PEG as a hydrophilic coating polymer has recently shown some demerits. It was reported that the PEG chain stimulated hypersensitivity reactions. It was also found that it was not biodegradable, which hurdles renal excretion and leads to the deposition of hepatic and lysosomes of healthy tissues [266]. Another drawback was an immune response by detecting anti-PEG antibodies (reaching 25% of the population) [267]. Additionally, repeated intravenous administration of PEGylated formulations was reported to induce splenic anti-PEG IgM, which deprived the formulation of its advantageous long circulation by “accelerated blood clearance” phenomenon [268].

Furthermore, scale-up production and continuous manufacturing are other challenges facing polymeric nanoparticles [269].

## 8. Recent Advances

Lipid–polymer hybrid nanocarriers have been recently used as alternatives to polymeric nanoparticles. They are hybrid systems containing both lipid and polymeric components. So, the system collects the benefits of the lipid-based nanoparticles, particularly biomimetic and biocompatibility natures, along with the merits of polymeric nanoparticles, particularly controlled release behavior and targeting [270]. The outer shell is represented by the lipid, while the inner core is represented by the polymer. The system offers a high payload [271], controlled release pattern, non-immunogenicity [272], and capability of surface decoration by desired ligands [273]. Quirós-Fallas and coworkers developed curcumin-loaded hybrid lipid polymeric nanoparticles and demonstrated efficient antioxidant activity, immune–stimulant response, and cytotoxic effect against both gastric and colon cancer cell lines [274]. Naziris and coworkers developed curcumin-loaded mixed lipid–polymer nanoparticles and found higher interaction of the system with bacterial membrane proteins leading to an efficient antibacterial effect [275]. Abosabaa and coworkers developed green tea extract-loaded hybrid chitosan–lipid nanoparticles. Results showed higher skin permeability and deposition of the system when compared to the solution. In addition, the formula showed promising anti-cellulite activity and reduction of adipocyte perimeter and fat layer thickness of the skin [276]. Patel and coworkers developed hybrid chitosan–lipid nanoparticles co-loaded with mycophenolic acid and quercetin. In vivo results showed higher half-life (2.17-fold when compared to free mycophenolic acid) and promising anti-breast cancer activity [277].

Stimuli-responsive polymeric nanosystem is another system that has garnered much attention [278]. The system has been efficiently utilized in controlled drug delivery, particularly in cancer therapy [279]. However, the stimuli may be internal (patho-physiological/patho-chemical condition), such as pH [280], reduced glutathione [281], and protease [282], or external (physical), such as light [283], temperature [284], and ultrasound [285]. Regarding internal stimuli, tumor tissues are manifested by a slightly low pH [286], a high amount of reduced glutathione [287], and the overexpression of protease enzyme [288]. On the other hand, near-infrared light [289], ultrasound [290], and temperature [291] were reported to control the release from some polymeric materials. Chen and coworkers developed dual hydrogen peroxide (as an extracellular trigger)/glutathione (as an intracellular trigger)-responsive thioketal polymeric nanoparticles for programmable paclitaxel release at the tumor site [292]. Among stimuli-responsive polymeric nanosystems, nanogel-based polymeric nanoparticles have established a high capability to uptake water in 3D nano-polymeric networks. Modern findings have revealed that the disruption of nanogels can be performed by using cross-linking agents sensible to temperature, light, and change in pH. Consequently, the loaded material can reach intracellular targets after the endocytosis of nanogel [293,294]. Howaili and coworkers developed curcumin-loaded dual thermo/pH-responsive plasmonic nanogel by grafting poly (N-isopropyl acrylamide) to chitosan with further incorporation into gold nanoparticles to provide simultaneous delivery and photothermal therapy of cancer. The results showed an enhanced chemotherapy efficacy when irradiated with near-infrared laser in vitro [295].

## 9. Perspectives and Conclusions

Throughout the whole world, attention has become increasingly directed toward the use of natural bioactive agents owing to their excellent therapeutic and preventive benefits. They possess several actions, such as antioxidant, anti-inflammatory, antimicrobial, and chemopreventive actions, which are not solely sufficient to provide to a patient. Thus, finding relevant formulation/technology to deliver them through different routes is the biggest challenge due to their drawbacks, such as poor solubility, poor bioavailability, and instability. At present, a wide range of polymeric nanoparticles (naturally or synthetically based) has been found to be relevant for the encapsulation or loading of different natural bioactive agents in order to enhance their bioavailability and therapeutic efficacy. Though polymeric nanoparticles were recognized by some researchers for their toxicity (particularly in high concentrations), surface functionalization is considered a key factor that can overcome such problems. In addition, using a lipid–polymer hybrid system and stimuli-responsive nanocarrier has opened a new horizon for more safe and site-specific drug release. So, the future of such new polymeric nanoparticles will enable scientists to provide safe and smart delivery of natural bioactive agents. Overall, the use of natural bioactive agent-loaded polymeric nanoparticles is a promising strategy, provided that the platform used fulfills the therapeutic and safety aspects.

## Figures and Tables

**Figure 1 polymers-15-01123-f001:**
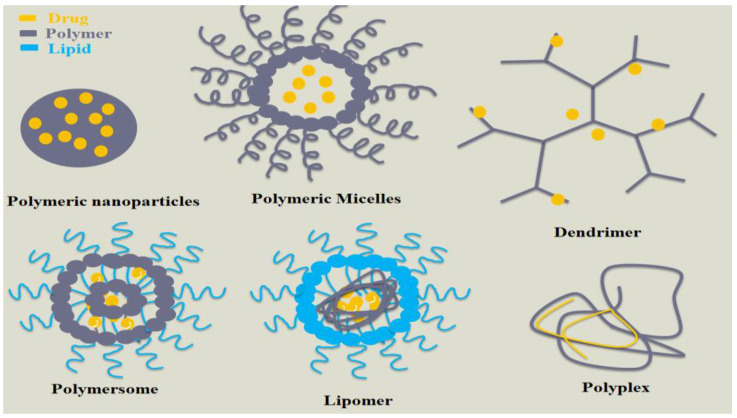
Different polymeric-based nanocarrier systems.

**Figure 2 polymers-15-01123-f002:**
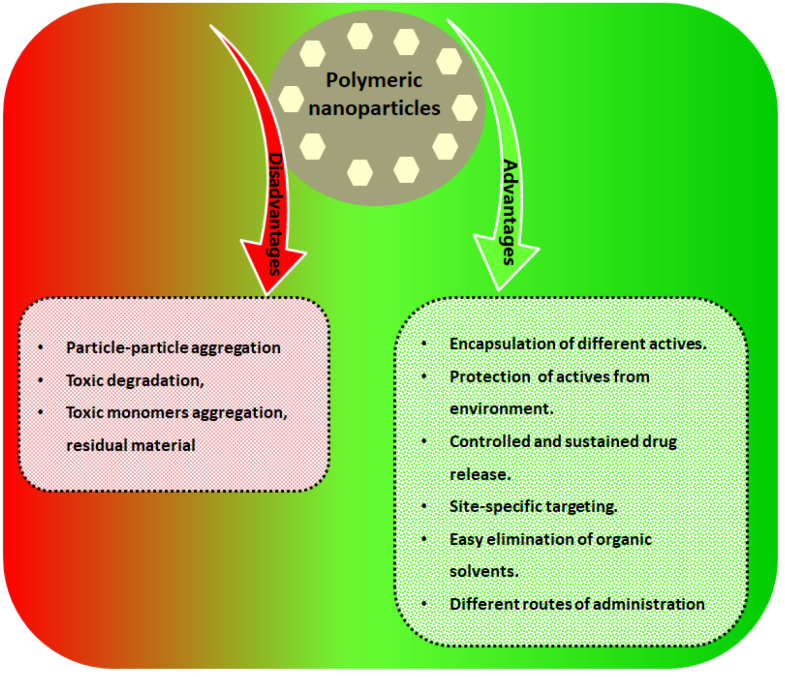
Advantages and disadvantages of polymeric nanoparticles.

**Figure 3 polymers-15-01123-f003:**
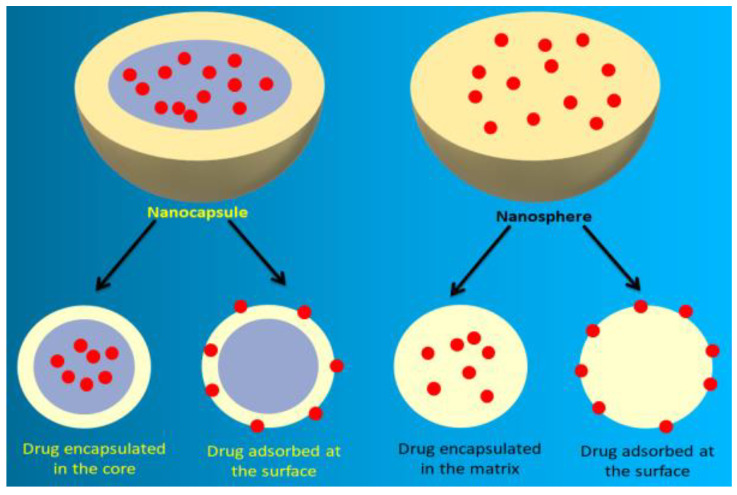
Classification of polymeric nanoparticles according to structural organization.

**Figure 4 polymers-15-01123-f004:**
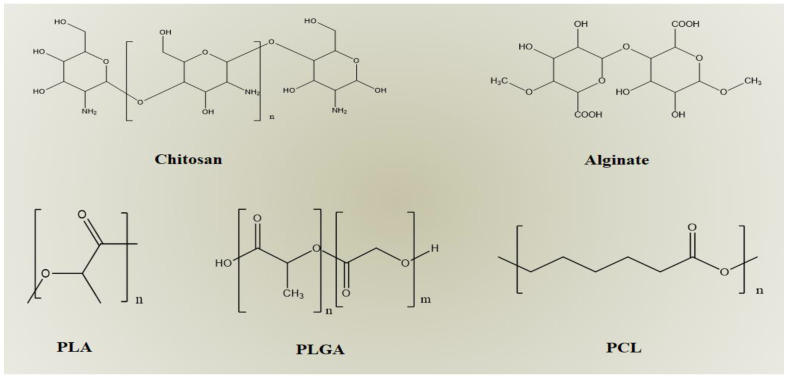
Chemical structures of different polymeric materials used in fabrication of polymeric nanoparticles.

**Figure 5 polymers-15-01123-f005:**
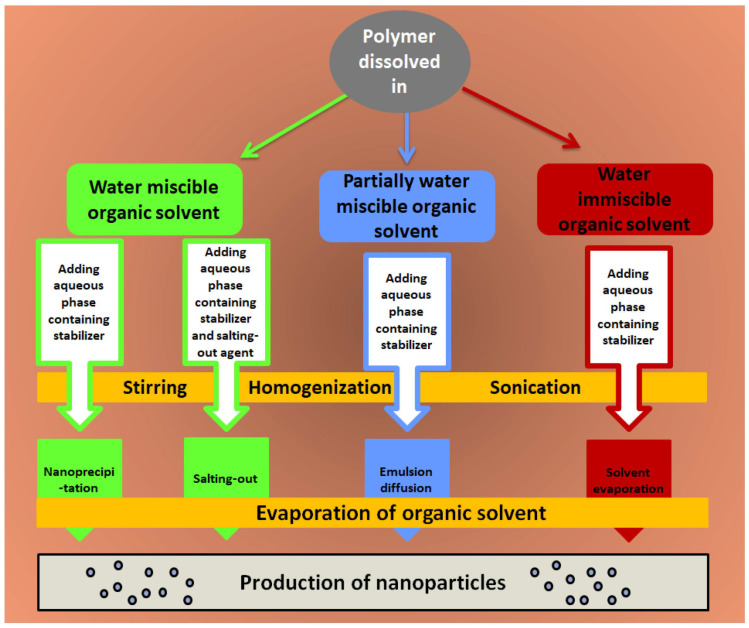
Schematic representation of frequently used techniques used to fabricate synthetic-based polymeric nanoparticles.

**Figure 6 polymers-15-01123-f006:**
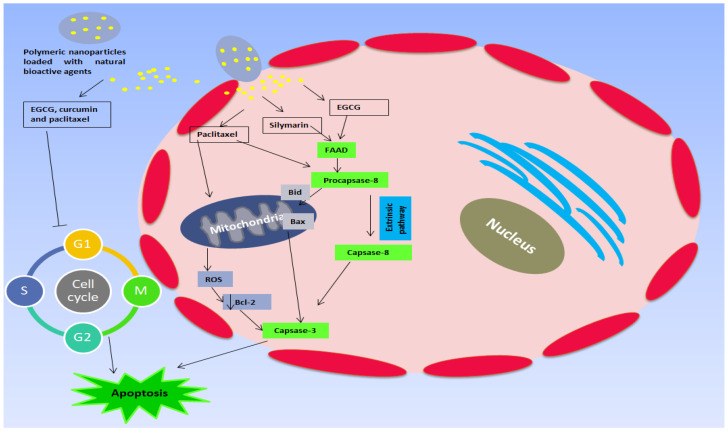
Schematic representation showing the mechanisms by which natural bioactive agents can act as anticancer agents after release from polymeric nanoparticles.

**Table 1 polymers-15-01123-t001:** Examples of recent studies utilizing natural polymer-based polymeric nanoparticles for loading natural bioactive agents.

Polymeric Composition		Loaded Natural Bioactive Agent	Research Outcomes	Reference
Natural polymers	Chitosan/chitosan derivatives	Quercetin	Efficient uptake by HaCaT cells. Improvement of percutaneous absorption and skin retention. Enhancement of anti-UVB effect.	[11]
			Effective antimicrobial and antiadhesion effects in multidrug-resistant isolates (*E. coli* and *S. aureus*).	[12]
			Significant cytotoxic effect against human breast tumor (MCF-7) and human lung tumor (A549).	[13]
			Improvement of wound healing by modulation of cytokines and growth factors involved in inflammatory and proliferative phases of wound healing.	[14]
		Curcumin	Improved antimicrobial activities against *C. glabrata* and *A. niger.*	[15]
			Improved anticancer activity against Vero cell line.	[16]
			Improved neuroprotective effect of curcumin and curcumin-loaded chitosan nanoparticles against stress-induced neurobehavioral and neurochemical deficits and protection against stress-associated gastric ulcer.	[17]
			Improved mucoadhesion, cellular uptake, and cytotoxicity against Caco-2 cells. Improved antioxidant and anti-inflammatory effects in activated RAW264.7 cells.	[18]
			Higher suppression of hepatocellular carcinoma growth in murine xenograft models and inhibited tumor angiogenesis when compared to free curcumin.	[19]
		Resveratrol	Larger AUC0-6 (6.44-fold) and higher mean residence time in plasma (2.458-fold) of the optimized formulation when compared to of resveratrol solution.	[20]
			Higher solubility and stability of resveratrol-loaded nanoparticles.	[21]
			Negligible in vitro release of resveratrol in simulated gastric. Sustained release in simulated intestinal conditions. Higher relative bioavailability of resveratrol.	[22]
			Improved anticancer activity against HepG2 cells.	[23]
			Significant improvement in the in vivo permeation of resveratrol. Improvement in the human stratum corneum penetration of resveratrol after topical application.	[24]
		Silymarin	Significantly improved depressive-like behaviors. Significantly reduced levels of malondialdehyde (MDA) and expression of interleukin-6 (IL-6) and tumor necrosis factor-α (TNF-α). Significant increase in the activities of superoxide dismutase (SOD), catalase (CAT), glutathione peroxidase, glutathione reductase, and glutathione (GSH) levels in I/R brain.	[25]
		Saponin	Nanosaponin showed higher uptake and specific toxicity on PC3 and KB cell lines.	[26]
		Paclitaxel	Safe and High loading efficiency and encapsulation efficiency.	[27]
		Essential oils	Improved antifungal activity against *Aspergillus versicolor*, *A. niger,* and *Fusarium oxysporum*.	[28]
			Improved antioxidant activity Improved antibacterial activity against *L. monocytogenes* and *S. aureus*.	[29]
	Alginate	Quercetin	Effective down-regulating the inflammation-related gene expression of tumor necrosis factor-α, interleukin-6, inducible nitric oxide synthase, and monocyte chemotactic protein-1.	[30]
		Curcumin	Prolonged release of curcumin. Enhanced collagenesis and increased number of fibroblasts. Improved wound healing efficacy.	[31]
			Amelioration of colonic inflammation and tissue damage by inhibiting the TLR4 expression in colonic epithelial cells. Reducing the transcription and expression of the pro-inflammation cytokines downstream.	[32]
	Gelatin	Resveratrol	Greater ROS generation, DNA damage, and apoptotic incidence. Higher bioavailability and a longer half-life than free resveratrol.	[33]
			High loading efficiency and superior efficacy in NCI-H460 cells.	[34]
	Albumin	Silymarin	Enhancement in the viability of hepatocytes in vitro against both APAP and LPS/D-GalN-induced hepatocyte damage. Improvement of antioxidant effects against intracellular oxidative stress.	[35]
		Curcumin	Enhanced cytotoxicity on triple-negative human breast cancer cells (MDA-MB-231) compared to free curcumin.	[36]
		Resveratrol	Potent activity against oxidative stress-based diseases.	[37]
		Paclitaxel/resveratrol	High encapsulation efficiency. Sustained drug release profiles.	[38]

**Table 2 polymers-15-01123-t002:** Examples of recent studies utilizing synthetic polymer-based polymeric nanoparticles for loading natural bioactive agents.

Polymeric Composition		Loaded Natural Bioactive Agent	Research Outcomes	Reference
Synthetic polymers	PLA	Paclitaxel	Suppression of the tumor growth in mice. Better tumor suppression trend with conjugated resveratrol nanoparticles with no side effects.	[61]
			Significant therapeutic improvement in arthritis proved by measuring rats’ knee diameter as well as the tumor necrosis factor-alpha (TNF-α).	[62]
			Effective targeting of folate-decorated paclitaxel-loaded copolymer nanoparticles on cancer cells both in vitro and in vivo.	[63]
			Improved anti-tumoral activity of paclitaxel when compared to the commercial paclitaxel formulation Taxol^®^.	[64]
			Enhanced cellular uptake in both human umbilical vein endothelial cells and rat C6 glioma cells. Increased cytotoxicity and improved penetration and growth inhibition in avascular C6 glioma spheroids.	[65]
		Quercetin	Enhanced anticancer efficacy in terms of its sustained release kinetics revealing novel vehicle for the treatment of cancer.	[66]
			Initial burst release followed by slow and sustained release. High antioxidant activity.	[67]
			Good antibacterial effects against *Staphylococcus aureus* (*S. aureus*), *Escherichia coli* (*E. coli*), and *Klebsiella pneumoniae* (*K. pneumoniae*).	[68]
		Curcumin	Amelioration of the negative changes in diabetes with a more pronounced treated effect than free curcumin.	[69]
			Cytocompatibility and suppressed production of TNF-α.	[70]
		Carvacrol	Enhanced in vitro antimicrobial activity against *Escherichia coli*, *Listeria monocytogenes*, *Salmonella enterica,* and *Staphylococcus aureus*.	[71]
	PLGA	Grape extracts	Sustained release with good stability in GIT fluids.	[72]
		Curcumin and docetaxel	Significant increase in cytotoxic activity. Significantly improved brain penetration.	[73]
		Curcumin	Higher percentage of curcumin internalization in cells, leading to enhanced cancer cell killing with targeted nanoparticles.	[74]
			Increased curcumin means half-life and high Cmax. Improved oral bioavailability.	[75]
		Quercetin	Production of thermosensitive gel as cost-effective option for burn wound therapy.	[76]
			Improved anti-oxidant activity.	[77]
			Significant blockage of UVB irradiation-induced COX-2 up-expression and NF-kB activation in Hacat cell line.	[78]
			Inhibition of the neurotoxicity of Zn^2+^-Aβ42 system and enhancement of the viability of neuron cells.	[79]
		Plant extract	Decreased level in the inflammation-related gene expression and DNA fragmentation level. High survival rate of post-damaged corneal epithelial cells. Enhanced corneal wound healing and decreased inflammation.	[80]
	PCL	Essential oil	Powerful antibacterial and antifungal activities that may be utilized as dermal patches for acne treatment.	[81]
			Prompt antioxidant activity. Inhibition of *Staphylococcus aureus* and *Escherichia coli* growth.	[82]
		Curcumin	Increase in the cell population of Sub G1 and apoptosis. Decreased tumor cell proliferation and angiogenesis.	[83]
			Enhanced antibacterial efficacy against both Gram-positive and Gram-negative strains. Proficient scar-less wound healing activity within 21 days with marked re-epithelization of tissue, collagen deposition, and formation of granulation tissue.	[84]
	PAMAM	Paclitaxel and curcumin	High encapsulation efficiency. Improved bioavailability and bioactivity against skin cancer.	[85]
	Methacrylate	Epigallocatechin gallate	Enhanced antimicrobial activity of EGCG-attached polymer that can be used as an alternative strategy to preclude microbial colonization.	[86]
		Curcumin	Enhanced tumor cell regression activity when compared to free curcumin. Significant reduction in G0/G1 cells. High biocompatibility.	[87]
		Silymarin	High capability to reverse the induced liver fibrosis and decrease serum TNF-α, serum TGF-β1, and hepatic hydroxyproline. Down-regulation in the expression of different hepatic biomarkers. Very little collagen of livers in extracellular matrix and restored hepatic architecture.	[88]

## Data Availability

Not applicable.

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
