# Peer review of "Polymeric Nanoparticles for Delivery of Natural Bioactive Agents: Recent Advances and Challenges"

_polymers, 2023, doi:10.3390/polym15051123_

Round 1
Reviewer 1 Report
The authors of the paper wrote mainly clearly and concisely. It is easy for someone who works in theory to understand the experimental procedures. I have a few point that I feel need to be addressed.
1) Figure 1 looks a bit sloppy and hand-drawn . Make the figure more aesthetically pleasing and professional.
2) Table 1 is big and multi-page. This is confusing to the reader. Perhaps break down the table into smaller tables. One table should not be multi-page.
3) Why are lines 65-100 and line 403 different fonts. Have the text one consistent font/size.
4) Make 2-dimensional drawings on monomers and bioactive agents the polymers carry. This will help chemists and biologist understand the polymers and the agents they carry.
5) No paragraph break in lines 392-393 necessary. It is in amiddle of a sentence.
6) Have you looked into flavonoids and polyphenols as a way to inhibit metabolite crystal growth formation? One example is in a paper by S. Shaham et al. "Differential Inhibition of Metabolite Amolyoid Formation by Generic Fibrillation-Modifying Polyphenols."
7) Break down lines 470-511 and 636-686 into smaller paragraphs. The reader will have an easier time to follow.
8) Check the punctuation for etc. in line 732.
Author Response
Reviewer 1
Comments and Suggestions for Authors
The authors of the paper wrote mainly clearly and concisely. It is easy for someone who works in theory to understand the experimental procedures. I have a few point that I feel need to be addressed.
1) Figure 1 looks a bit sloppy and hand-drawn. Make the figure more aesthetically pleasing and professional.
Authors’ reply:
Authors are thankful to the learned referee for the comment. We have replaced the figure by more aesthetically pleasing one.
2) Table 1 is big and multi-page. This is confusing to the reader. Perhaps break down the table into smaller tables. One table should not be multi-page.
Authors’ reply:
We thank the learned referee for the comment. We have separated the Table 1 into two tables in revised version (Table 1 for natural polymers-based nanoparticles and Table 2 for synthetic polymers-based nanoparticles).
3) Why are lines 65-100 and line 403 different fonts. Have the text one consistent font/size.
Authors’ reply:
Authors are thankful to the learned referee for the comment. We have made all text as one sized font style and size.
4) Make 2-dimensional drawings on monomers and bioactive agents the polymers carry. This will help chemists and biologist understand the polymers and the agents they carry.
Authors’ reply:
We express our gratitude to the learned referee for providing this valuable suggestion. We have added 2-D of polymeric monomers as new image in the revised version (Figure 4). The structures have been drawn using Chemdraw software.
5) No paragraph break in lines 392-393 necessary. It is in amiddle of a sentence.
Authors’ reply:
Ok. It has been corrected.
6) Have you looked into flavonoids and polyphenols as a way to inhibit metabolite crystal growth formation? One example is in a paper by S. Shaham et al. "Differential Inhibition of Metabolite Amolyoid Formation by Generic Fibrillation-Modifying Polyphenols."
Authors’ reply:
Ok. The reported pharmacological effect of polyphenols has been added in the manuscript and relevant reference has been cited.
7) Break down lines 470-511 and 636-686 into smaller paragraphs. The reader will have an easier time to follow.
Authors’ reply:
Ok. They have been broken down into smaller paragraphs.
8) Check the punctuation for etc. in line 732.
Authors’ reply:
Ok. It has been corrected.

Reviewer 2 Report
The manuscript entitled “Polymeric nanoparticles for delivery of natural bioactive agents: Recent advances and challenges” provides an in-depth overview of the use of polymeric nanoparticles as therapeutic delivery platforms, with discussions on frequently used polymeric nanoparticles and their fabrication strategies, the roles of bioactive agents used in these systems in the literature, as well discussions regarding the use of surface decoration and polymer functionalization to address the drawbacks of these platforms.
Overall, the manuscript is very informative, well organized, and well-supported by relevant literature references. I believe this will be a helpful resource for researchers pursuing this area of research. The writing could benefit from polishing, as there are a number of places with incorrect grammar or where the clarity could be improved. I believe that this manuscript is suitable for publication pending minor revisions, and some suggestions in this regard are outlined below.
Line 18: The phrase “so proficient delivery of various” is unclear.
Line 35: “and microbial threatens” can be changed to “and microbial threats”.
Lines 50-51: “have considered as promising platform” can be changed to “have been considered as promising platforms”.
Line 59: “particle size range from” can be changed to “particle sizes ranging from”.
Lines 64-65: The phrase “polymers paid the concern due to” is unclear.
Line 78: “most of system drawbacks” can be changed to “most of the drawbacks of these systems”.
Line 80: “composed polymer(s),” can be changed to “composed of polymer(s),”.
Line 85: “as cornerstone” can be changed to “as cornerstones” or “as the cornerstone”.
Lines 91-92: References may be needed for the last sentence in the first paragraph of section 2.
Lines 97 and 99: “active might be” can be changed to “active agents might be” or “bioactive agents might be”.
Line 120: “is using very less amount of organic solvent” can be changed to “is that they use much smaller amounts of organic solvent”.
Line 136: “can carried out” can be changed to “can be carried out”.
Lines 156-157: “it has been gained the attention as drug delivery platform in pharmaceutical field” can be changed to “it has gained the attention as a drug delivery platform in the pharmaceutical field” or “it has attracted attention as a drug delivery platform in the pharmaceutical field”.
Lines 166-167: A reference may be needed for the statement describing how gelatin is obtained.
Line 175: “It is” can be changed to “It is a”.
Line 190: “is aliphatic polyester” can be changed to “is an aliphatic polyester”.
Line 206: the phrase “it has been tried for using in” is unclear.
Lines 222-224: References may be needed for the first sentence in the section describing PAMAM.
Line 225: “dedrimers was” can be changed to “dendrimers were”.
Line 242: “Another problems associated” can be changed to “Other problems associated”.
Line 248: “Establishing suitable drug delivery system” can be changed to “Establishing a suitable drug delivery system”.
Line 293: “polymeric nanoparticles acts as encouraging strategy to” may be changed to “polymeric nanoparticles may provide an effective way to”.
Lines 327-328: the sentence “A plethora of experimental data has clearly reported wide range of therapeutic activities” is unclear.
Line 361: Clarification may be needed to indicate whether or not the authors mentioned in the sentence beginning with “Authors studied different factors” are the same authors of the study mentioned in the previous sentence (reference [146]).
Lines 392-393: There seems to be a line spacing or formatting issue here.
Line 432: “It is existed” can be changed to “It exists”.
Line 513: “seeds which consisting of” can possibly be changed to “seeds which consist of”.
Line 517: “as chemopreventive” can be changed to “as a chemopreventive”.
Line 518: “nnaoparticles” can be changed to “nanoparticles”.
Line 521: “Azadpour coworkers” can be changed to “Azadpour and coworkers”.
Line 554: “by emulsion solvent diffusion method” can be changed to “by an emulsion solvent diffusion method”.
Line 589: “Dual-targeting approach” can be changed to “A dual-targeting approach”.
Line 609: “to to achieve” can be changed to “to achieve”.
Line 627: “should the following criteria” can be changed to “should take the following criteria”.
Lines 637-638: The phrase “In the first one, reactive group is one-side attached to the nanoparticle surface and” seems to be unclear.
Line 659: “with hydrophilic polymer” can be changed to “with a hydrophilic polymer”.
Line 680: “Pluronics (Poloxamers) was reported to” can possibly be changed to “Pluronics (Poloxamers) were reported to”.
Line 697: “is aroused form” can possibly be changed to “is attributable to” or “results from”.
Line 732: “chemopreventive ..etc which are not solely” can be changed to “chemopreventive, etc., which are not solely” or ““chemopreventive, and so forth, which are not solely”.
Line 735: “At present, wide range of” can be changed to “At present, a wide range of”.
Author Response
Reviewer 2
Comments and Suggestions for Authors
The manuscript entitled “Polymeric nanoparticles for delivery of natural bioactive agents: Recent advances and challenges” provides an in-depth overview of the use of polymeric nanoparticles as therapeutic delivery platforms, with discussions on frequently used polymeric nanoparticles and their fabrication strategies, the roles of bioactive agents used in these systems in the literature, as well discussions regarding the use of surface decoration and polymer functionalization to address the drawbacks of these platforms.
Overall, the manuscript is very informative, well organized, and well-supported by relevant literature references. I believe this will be a helpful resource for researchers pursuing this area of research. The writing could benefit from polishing, as there are a number of places with incorrect grammar or where the clarity could be improved. I believe that this manuscript is suitable for publication pending minor revisions, and some suggestions in this regard are outlined below.
Line 18: The phrase “so proficient delivery of various” is unclear.
Authors’ reply:
Authors are thankful to the learned referee for the comment. The statement has been paraphrased to be more clear.
Line 35: “and microbial threatens” can be changed to “and microbial threats”.
Authors’ reply:
Ok. It has been corrected.
Lines 50-51: “have considered as promising platform” can be changed to “have been considered as promising platforms”.
Authors’ reply:
Ok. It has been corrected.
Line 59: “particle size range from” can be changed to “particle sizes ranging from”.
Authors’ reply:
Ok. It has been corrected.
Lines 64-65: The phrase “polymers paid the concern due to” is unclear.
Authors’ reply:
Authors are thankful to the learned referee for the comment. The statement has been corrected.
Line 78: “most of system drawbacks” can be changed to “most of the drawbacks of these systems”.
Authors’ reply:
Ok. It has been corrected.
Line 80: “composed polymer(s),” can be changed to “composed of polymer(s),”.
Authors’ reply:
Ok. It has been corrected.
Line 85: “as cornerstone” can be changed to “as cornerstones” or “as the cornerstone”.
Authors’ reply:
Ok. It has been corrected.
Lines 91-92: References may be needed for the last sentence in the first paragraph of section 2.
Authors’ reply:
Ok. Relevant reference has been added.
Lines 97 and 99: “active might be” can be changed to “active agents might be” or “bioactive agents might be”.
Authors’ reply:
Ok. It has been corrected.
Line 120: “is using very less amount of organic solvent” can be changed to “is that they use much smaller amounts of organic solvent”.
Authors’ reply:
Ok. It has been corrected.
Line 136: “can carried out” can be changed to “can be carried out”.
Authors’ reply:
Ok. It has been corrected.
Lines 156-157: “it has been gained the attention as drug delivery platform in pharmaceutical field” can be changed to “it has gained the attention as a drug delivery platform in the pharmaceutical field” or “it has attracted attention as a drug delivery platform in the pharmaceutical field”.
Authors’ reply:
Ok. It has been corrected.
Lines 166-167: A reference may be needed for the statement describing how gelatin is obtained.
Authors’ reply:
Ok. Relevant reference has been added.
Line 175: “It is” can be changed to “It is a”.
Authors’ reply:
Ok. It has been corrected.
Line 190: “is aliphatic polyester” can be changed to “is an aliphatic polyester”.
Authors’ reply:
Ok. It has been corrected.
Line 206: the phrase “it has been tried for using in” is unclear.
- The statement has been paraphrased to be more clear.
Lines 222-224: References may be needed for the first sentence in the section describing PAMAM.
Authors’ reply:
Ok. Relevant reference has been added.
Line 225: “dedrimers was” can be changed to “dendrimers were”.
Authors’ reply:
Ok. It has been corrected.
Line 242: “Another problems associated” can be changed to “Other problems associated”.
Authors’ reply:
Ok. It has been corrected.
Line 248: “Establishing suitable drug delivery system” can be changed to “Establishing a suitable drug delivery system”.
Authors’ reply:
Ok. It has been corrected.
Line 293: “polymeric nanoparticles acts as encouraging strategy to” may be changed to “polymeric nanoparticles may provide an effective way to”.
Authors’ reply:
Ok. It has been corrected.
Lines 327-328: the sentence “A plethora of experimental data has clearly reported wide range of therapeutic activities” is unclear.
Authors’ reply:
- The statement has been paraphrased to be more clear.
Line 361: Clarification may be needed to indicate whether or not the authors mentioned in the sentence beginning with “Authors studied different factors” are the same authors of the study mentioned in the previous sentence (reference [146]).
Authors’ reply:
- Yes, the authors are the same mentioned in reference 146. So, we have added the reference at the end to be more clear.
Lines 392-393: There seems to be a line spacing or formatting issue here.
Authors’ reply:
- It has been changed.
Line 432: “It is existed” can be changed to “It exists”.
Authors’ reply:
Ok. It has been corrected.
Line 513: “seeds which consisting of” can possibly be changed to “seeds which consist of”.
Authors’ reply:
Ok. It has been corrected.
Line 517: “as chemopreventive” can be changed to “as a chemopreventive”.
Authors’ reply:
Ok. It has been corrected.
Line 518: “nnaoparticles” can be changed to “nanoparticles”.
Authors’ reply:
Ok. It has been corrected.
Line 521: “Azadpour coworkers” can be changed to “Azadpour and coworkers”.
Authors’ reply:
Ok. It has been corrected.
Line 554: “by emulsion solvent diffusion method” can be changed to “by an emulsion solvent diffusion method”.
Authors’ reply:
Ok. It has been corrected.
Line 589: “Dual-targeting approach” can be changed to “A dual-targeting approach”.
Authors’ reply:
Ok. It has been corrected.
Line 609: “to to achieve” can be changed to “to achieve”.
Authors’ reply:
Ok. It has been corrected.
Line 627: “should the following criteria” can be changed to “should take the following criteria”.
Authors’ reply:
Ok. It has been corrected.
Lines 637-638: The phrase “In the first one, reactive group is one-side attached to the nanoparticle surface and” seems to be unclear.
Authors’ reply:
- The statement has been paraphrased to be more clear.
Line 659: “with hydrophilic polymer” can be changed to “with a hydrophilic polymer”.
Authors’ reply:
Ok. It has been corrected.
Line 680: “Pluronics (Poloxamers) was reported to” can possibly be changed to “Pluronics (Poloxamers) were reported to”.
Authors’ reply:
Ok. It has been corrected.
Line 697: “is aroused form” can possibly be changed to “is attributable to” or “results from”.
Authors’ reply:
Ok. It has been corrected.
Line 732: “chemopreventive ..etc which are not solely” can be changed to “chemopreventive, etc., which are not solely” or ““chemopreventive, and so forth, which are not solely”.
Authors’ reply:
Ok. It has been corrected.
Line 735: “At present, wide range of” can be changed to “At present, a wide range of”.
Authors’ reply:
Ok. It has been corrected.

Reviewer 3 Report
This review is not well organized and outlined, so authors must address the following comments before a positive decision can be made.
The category of polymeric nanoparticles should be revised, since the authors miss some important types, including liposome, emulsion etc.
The introduction should be revised to justify the difference of this review from the others.
The challenges for this application are too limited, and should be strengthened and considered more.
Some schematic illustrations on the examples should be included to improve the quality of this review.
The outline is discrete and not reasonable to follow. Authors must address that.
The latest advances on this field should be included and summarized.
The clinical advances should be discussed and mentioned.
Author Response
Reviewer 3
This review is not well organized and outlined, so authors must address the following comments before a positive decision can be made.
The category of polymeric nanoparticles should be revised, since the authors miss some important types, including liposome, emulsion etc.
Authors’ reply:
We appreciate the learned referee for this valuable comment. In our work, we focused on polymeric based nanoparticles and not on any other type of nanoparticles. In general, nanoparticles are classified into different categories depending on the base of classification. For example, they may be classified into organic and inorganic nanoparticles (please see references 1-4). Organic class includes polymeric nanoparticles, lipid nanoparticles, micelles and dendrimers. In another classification, they are classified into Conventional Surfactants, Lipids (such as microemulsion, SLN and liposomes), Polymers, and Inorganic materials (please see reference no. 5). Furthermore and according to integrity of the particles, they are classified into nonrigid nanoparticles (such as liposomes and SLN) and rigid nanoparticles (such as polymeric nanoparticles (please see reference no. 6). So, in any classification liposomes and emulsions are not considered as polymeric based nanoparticles.
- Flavia Laffleur, Val´erie Keckeis, 2020. Advances in drug delivery systems: Work in progress still needed? Int. J. Pharm. 590 (2020) 119912
2.Rizvi, S.A.A., Saleh, A.M., 2018. Applications of nanoparticle systems in drug delivery technology. Saudi Pharm J. 26 (1), 64–70. https://doi.org/10.1016/j. jsps.2017.10.012.
3.Thakor, A.S., Gambhir, S.S., 2013. Nanooncology: The future of cancer diagnosis and therapy. CA Cancer J. Clin. 63 (6), 395–418. https://doi.org/10.3322/caac.21199.
4.Martinelli, C., Pucci, C., Ciofani, G., 2019. Nanostructured carriers as innovative tools for cancer diagnosis and therapy. APL Bioeng. 3 (1) https://doi.org/10.1063/ 1.5079943.
- Kevin Letchford, Helen Burt, 2007. A review of the formation and classification of amphiphilic
block copolymer nanoparticulate structures: micelles, nanospheres, nanocapsules and polymersomes. European Journal of Pharmaceutics and Biopharmaceutics 65 (2007) 259–269
6.Rajashekar Kammari, Nandita G. Das, Sudip K. Das, 2017. Chapter 6 - Nanoparticulate Systems for Therapeutic and Diagnostic Applications, Editor(s): Ashim K. Mitra, Kishore Cholkar, Abhirup Mandal, In Micro and Nano Technologies, Emerging Nanotechnologies for Diagnostics, Drug Delivery and Medical Devices, Elsevier, Pages 105-144.
The introduction should be revised to justify the difference of this review from the others.
Authors’ reply:
We express our gratitude to the learned referee for providing this valuable suggestion. We have added more details in the “Introduction”-section to illustrate the importance of the current work. All changes have been yellow highlighted.
The challenges for this application are too limited, and should be strengthened and considered more.
Authors’ reply:
We appreciate the learned referee for this valuable comment. More details have been added to “Challenges”-section. All changes have been yellow highlighted.
Some schematic illustrations on the examples should be included to improve the quality of this review.
Authors’ reply:
We are thankful to the learned referee for the valuable suggestion. We have added one more schematic diagram showing the anticancer mechanisms of natural bioactive agents after release from polymeric nanoparticles (Figure 6).
The outline is discrete and not reasonable to follow. Authors must address that.
Authors’ reply:
We appreciate the learned referee for this valuable comment. The outline has been changed and harmonized.
The latest advances on this field should be included and summarized.
Authors’ reply:
We appreciate the learned referee for this valuable comment. We have added new section entitled “Recent advances” to stand up for the latest developments in this regard. All changes have been yellow highlighted.
The clinical advances should be discussed and mentioned.
Authors’ reply:
We express our gratitude to the learned referee for providing this valuable suggestion. We have added new section entitled “Clinical trials” to stand up for the latest developments in this regard. All changes have been yellow highlighted.

Round 2
Reviewer 3 Report
I am satisfied with this revision and authors' response.